# RETVec: Resilient and Efficient Text Vectorizer

**Elie Bursztein**
Google
elieb@google.com

**Marina Zhang**
Google
marinazh@google.com

**Owen Vallis**
Google
ovallis@google.com

**Xinyu Jia**
Google
jiaxinyu@google.com

**Alexey Kurakin**
Google
kurakin@google.com

## Abstract

This paper describes RETVec, an efficient, resilient, and multilingual text vectorizer designed for neural-based text processing. RETVec combines a novel character encoding with an optional small embedding model to embed words into a 256-dimensional vector space. The RETVec embedding model is pretrained using pair-wise metric learning to be robust against typos and character-level adversarial attacks. In this paper, we evaluate and compare RETVec to state-of-the-art vectorizers and word embeddings on popular model architectures and datasets. These comparisons demonstrate that RETVec leads to competitive, multilingual models that are significantly more resilient to typos and adversarial text attacks. RETVec is available under the Apache 2 license at https://github.com/google-research/retvec.

## 1 Introduction

Researchers have proposed many techniques for converting text into dense representations suitable for training neural networks, from the early use of character bi-grams and tri-grams to more modern semantic word embeddings and state-of-the-art subword vectorizers. Each of these approaches aims to mitigate issues caused by out-of-vocabulary (OOV) tokens, including intentional text perturbations such as adversarial attacks and unintentional ones such as typos. These approaches leverage techniques such as subword-level tokenization [17] and decomposing unknown words into n-grams [4].

However, we find that each of these approaches suffers from at least one of following drawbacks:

- They have insufficient resilience against typos and adversarial attacks [21].
- They require large dictionaries and embedding lookup tables.
- They exhibit poor performance on certain languages or in multilingual settings.

RETVec addresses these issues by combining a novel UTF-8 character encoder with an optional small model (230k parameters). This model projects encoded words into a 256 dimensional **syntactic metric embedding** as detailed in Section 3. RETVec embeddings are trained using pair-wise metric learning [30], ensuring that words containing typos are embedded close to the the original word.

RETVec does not require dataset pre-processing and does not have OOV tokens because it accepts all valid UTF-8 characters. RETVec's embedding model is trained on a word dataset with more than 157 languages. RETVec is also space-efficient (<1MB) since it does not require a large embedding lookup table. This reduces model memory footprints, making RETVec-based models ideal candidates for on-device model deployment where storage, bandwidth, and memory resources are scarce. Through a series of extensive experiments on different model architectures, and on a wide range of datasets, we demonstrate that RETVec outperforms or is comparable to state-of-the-art vectorizers and word

37th Conference on Neural Information Processing Systems (NeurIPS 2023).

embeddings including BPE [26], SentencePiece [17], and fastText [4]. Overall, RETVec outperforms other vectorizers on text classification tasks by about 1%, while being up to 15% more resilient to typos at 20% word typo rate, and less susceptible to character-level adversarial attacks by over 10%.

Our paper makes the following contributions:

- We introduce RETVec, a resilient, efficient, and multilingual text vectorizer designed for neural-based text processing.
- We show that RETVec is faster and less memory intensive than other vectorizers on multi-core CPUs and on GPUs.
- We demonstrate that models trained with RETVec have slightly higher accuracy, greater resilience to typos, and significantly better resilience to adversarial attacks compared to models trained with the other vectorizers.
- We provide a TensorFlow implementation of RETVec, including its pre-trained models, alongside the code to reproduce our benchmarks under the Apache 2 license at `https://github.com/google-research/retvec`.

## 2  Background

**Text Vectorizers**   Text vectorizers play a major role in deep neural networks' (DNN) performance by ensuring that the input text is properly segmented into tokens (typically words or sub-words) and embedded into a dense, float-point representation for the model to use. The traditional approach for text vectorization splits texts into words using whitespace and punctuation and uses a word embedding lookup table to map each word into a dense vector. The embedding lookup table can be constructed using algorithms such as Word2Vec [20], GloVe [22] and fastText [4], or by randomly initializing an embedding table and training it as a part of the LLM pre-training process (like GPT-2 [2]). Embedding lookup tables are typically large and limited to known, in-vocabulary tokens which significantly reduces their representational capabilities in the presence of typos and adversarial attacks.

Subword tokenizers such as WordPiece [7], BPE [26] and SentencePiece [17]) split text into subwords to reduce vocabulary size and mitigate OOV issues. SentencePiece is used by many state-of-the-art text models [24, 5]. All of these vectorizers require a separate training phase to adapt to the targeted dataset and require shipping extra files with the models to be used at inference time.

**Pair-based learning**   The RETVec model is trained using pair-based metric learning. Pair-based learning aims to learn an embedding space, where embedded vectors of similar samples are encouraged to be closer together while dissimilar ones are pushed apart from each other [31]. This approach has been successfully applied to various tasks including image retrieval [12, 30, 27], face recognition [25], and zero-shot learning [34].

**Typos and Adversarial Attacks**   One challenge faced by text vectorizers is the existence of typographical errors that might not appear in the training dataset. At inference time, these typographical errors can lead the vectorizer to incorrectly output OOV ids for valid words. This behavior ultimately leads to lower accuracy, particularly for retrieval tasks where human queries are known to exhibit between 16% and 19% typos rate [35, 11]. In many anti-abuse settings such as spam classification, the typo rate used by attackers in attempt to evade defense systems is much higher [6]. It is also well-known that text models are vulnerable to different types of adversarial attacks [21]. Many character-level adversarial attacks, which can be viewed as the most severe form of intentional typos, have been proposed in the literature ( [18], [9], [23]).

## 3  RETVec

In this section, we describe how RETVec works. We start off by introducing its novel character encoding scheme, detail the design of its model, and finally describe its pre-training procedure.

### 3.1  RETVec Character Encoder

The RETVec character encoder, as shown in Figure 1, is composed of two layers: an `Integerizer` layer and a `Binarizer` layer. The character encoder enables RETVec to encode all UTF-8 characters

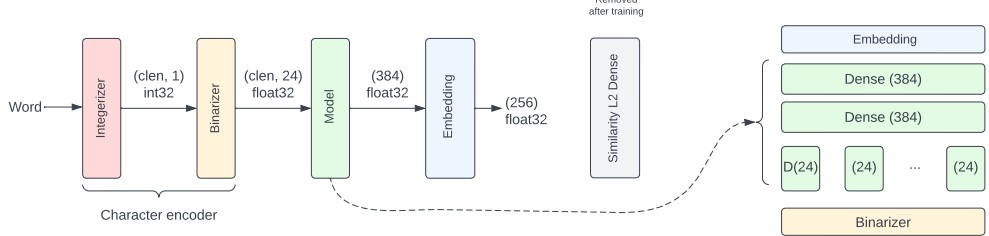

Figure 1: RETVec architecture overview - the output shape of each layer is in parenthesis. The `clen` indicates the number of characters used per word - 16 characters by default. The batch and word sequence length dimensions are omitted.

in an efficient and stateless manner. Encoding a word is achieved by first converting its characters into their UTF-8 codepoints (`Integerizer`) and then converting this integer representation into a flattened, binary (little-endian) representation (`Binarizer`). The `Binarizer` uses a compact 24-bit binary character representation to encode all valid UTF-8 characters. Additionally, RETVec outputs a fixed-length, dense word embedding using a maximum of 16 characters per word. In the ablation study (Section 10), we empirically find that 16 characters per word works well and increasing the max word length provides no further accuracy gains.

The `Integerizer` is similar to the one used in ByT5 [32] which also encodes characters into their UTF-8 codepoints. What makes the RETVec encoder unique is that the codepoints are first converted to their binary representation and then flattened to create the word representation. This novel binary representation is both easy to learn and significantly more compact than a one-hot representation. We tested many other approaches to convert the `Integerizer` output into a more compact form, including the use of a single float per character, but these more compact representations did not appear to be learnable. Additionally, a key design choice is to keep the `Integerizer` and the `Binarizer` separated for efficiency reasons: when using a remote accelerator, such as a TPU, it is more efficient to send the output of the `Integerizer` ($16\times$ int32) across the network rather than directly sending the binarization representation ($16 \times 24$ float32)).

## 3.2 RETVec Model Architecture

While the RETVec `Binarizer` provides an efficient word representation, it is not competitive with state-of-the-art vectorizers. To address this, we add a small model on top of the `Binarizer` output. This improves accuracy and enables RETVec to outperform other vectorizers (see Section 5). Additionally, this reduces the embedding size from 384 to 256 `float32`s and further improves resilience to adversarial attacks (see Section 7). The performance gains only incur a small increase in computational cost, making the use of the pre-trained model a worthwhile trade-off. To better understand what gains can be attributed to the character encoder, the remainder of the paper reports the performance for "RETVec with the model" as RETVec, and to understand what can be attributed to the model we report "RETVec without the model" as RETVec-raw.

The RETVec model, as visible in Figure 1, is composed of a dense projection layer, a flatten layer, a dense compression layer, and the embedding layer. The embedding layer is a dense layer with a `tanh` activation to scale the embedding values between 0 and 1, while the rest of the network uses `gelu` activations. Increasing capacity by adding additional layers or using more powerful architectures did not improve performance on our benchmarks (Section 10). Detailed ablation studies on loss hyperparameters, dropout rates, activation functions, and model capacity can be found in Appendix C.

## 3.3 RETVec Pre-training Procedure

**Dataset** The RETVec model is pre-trained on a typo-augmented version of the 157-language fastText words datasets [10]. The original words are extracted from the Common Crawl corpus. Our dataset then takes the set of words over the union of all 157 languages, yielding 88.8M unique

tokens. To help with generalization, we supplement this dataset by adding 8.88M (10%) randomly generated tokens, where the length of a token is randomized between 1 and 16. Each token consists of a random set of UTF-8 characters. Finally, we create 20 versions of each token (16 augmented and 4 non-augmented versions) such that 80% of the tokens in the training set are typo-augmented. At the end of the generation process we end-up with a combined 1.9B token dataset.

**Augmentations**   Token augmentation consists of randomly inserting up to 4 typos per token up to 25% of the token length. This is consistent with an observed maximum human error frequency of around 20% [11]. We use 22 distinct typo augmentations, which can be grouped into four categories: deletion, insertion, substitution, and transposition. For each token, we randomly select a target augmentation percentage between 0-25%, and for each augmentation step we randomly apply an augmentation from one of the four typo categories. The full list of augmentations used is reported in Appendix D.

**Training**   RETVec model is trained using pair-wise learning and uses the Multi-Similarity Loss [30] ($\alpha = 4$, $\beta = 40$, $\lambda = 0.5$, and $\epsilon = 0.1$). We construct example pairs by creating batches that always contain two variations of the same word. The model is trained for 500k steps with batch size = 1024, using Adam with max learning rate = 0.001, $\beta_1 = 0.9$, $\beta_2 = 0.999$, and cosine decaying the learning rate to 0.0001 during training. Full training hyperparameters can be found in Appendix E. We experimented with additional losses and other self-supervised tasks, reported in Appendix C, but none of them yielded better results.

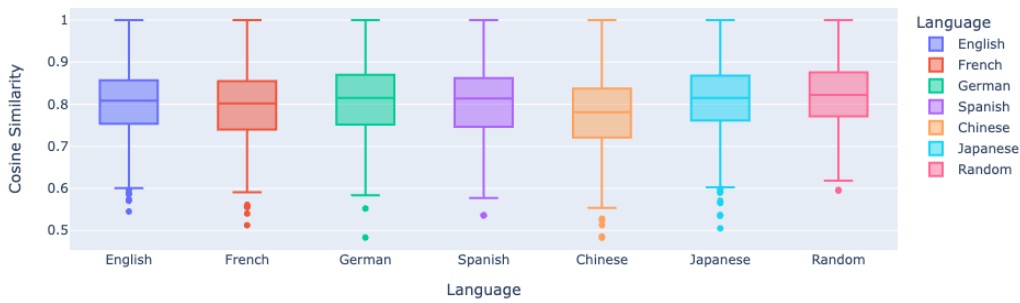

Figure 2: The cosine similarity distributions of RETVec embeddings for 1000 pairs of augmented and non-augmented versions of words, selected languages shown. 'Random' language refers to randomly-generated UTF-8 strings.

**Embedding Quality**   To visualize and evaluate the training quality and embedding consistency across languages, we randomly sample pairs of augmented and non-augmented words from select languages and compare the distribution of their cosine similarities. As shown in Figure 2, we observe that the mean and variance of the cosine similarity distributions are fairly consistent across various languages, which suggests that the model has learned near-uniformly across languages. The mean cosine similarity between word pairs in Chinese is slightly lower, which could be attributed to the fact that the average length of words is significantly shorter in Chinese. The distribution for random strings is also consistent with other languages' distributions, suggesting that RETVec is able to generalize to previously unseen words and embed words with rare UTF-8 characters meaningfully, which would greatly contribute towards its adversarial resilience. In Appendix J, we used the TensorBoard Embedding Projector to visualize RETVec embeddings in a 3D space and demonstrate that syntactically-similar words are clustered together, adding further evidence that RETVec is meaningfully projecting tokens into the embedding space.

## 4   Evaluation: Speed

In this section, we evaluate how quickly RETVec can process and vectorize datasets compared to the other commonly used vectorizers. We considered SentencePiece [17], BPE [26], a simple

| Name | Embedding Size | Vocabulary Size | Preprocessing Time | System Memory (GB) | Model Params | CPU | | | GPU | | |
|---|---|---|---|---|---|---|---|---|---|---|---|
| | | | | | | Wall time (s) | Usage | Core Sec | Wall (s) | Usage | Memory (GB) |
| SentencePiece | 256 | 32k | 749 | 10.9 | 8M | 877 | 124% | 1087 | 870 | 100% | 0.5 |
| BPE | 256 | 32k | 861 | 3.6 | 8M | 1062 | 169% | 1795 | 1050 | 100% | 0.5 |
| FastText | 300 | - | 420 | 43.9 | 0 | 845 | 52% | 439 | - | - | - |
| Whitespace | 256 | 32k | 132 | 1.7 | 8M | 496 | 126% | 625 | 396 | 100% | 0.5 |
| RETVec-raw | 384 | - | 0 | 1.9 | 0 | 225 | 451% | 1015 | 235 | 20% | 0.4 |
| RETVec | 256 | - | 0 | 2.1 | 230K | 570 | 840% | 4789 | 290 | 40% | 0.5 |

Table 1: Comparison of speed, preprocessing time, memory usage, and CPU/GPU usage for all benchmarked vectorizers when vectorizing the Multilingual Amazon Reviews training dataset.

| Name | Train Size | Test Size | # Classes | # Languages | OOV Words | Avg Sentence Length | Avg Word Length |
|---|---|---|---|---|---|---|---|
| AG News [33] | 120k | 7.6k | 4 | 1 | 9.9% | 45 | 4.4 |
| Yelp Reviews (Polarity) [33] | 560k | 38k | 2 | 1 | 17.4% | 160 | 3.7 |
| Multilingual Amazon Reviews (Polarity) [19] | 960k | 24k | 2 | 6 | 35.2% | 27 | 5.0 |
| MASSIVE (Intent Classification) [8] | 587k | 152k | 60 | 51 | 35.3% | 6 | 5.1 |

Table 2: List of datasets used for classification evaluation.

whitespace vectorizer, and fastText [4]. We use the HuggingFace implementation of BPE and the official SentencePiece implementation.

**Setup**   We use the Multilingual Amazon Reviews [19] dataset for the vectorization speed evaluations. The corpus is composed of 1.2M tokens – 200k tokens for 6 languages (English, French, German, Spanish, Chinese, Japanese). All the experiments are run on a standard Google Cloud VM with 16 CPU cores and a V100 Nvidia GPU. We disable the GPU visibility to perform the CPU measurements and use the `GNU time` command to measure the total wall time, CPU usage, and system memory. We use `nvidia-smi` to record GPU usage and memory. We report both the time it takes to adapt to the datasets and how long it takes to vectorize the entire dataset end-to-end, with and without GPU acceleration. We also report normalized CPU-core times as RETVec (and other vectorizers to a lesser extent) benefits from multi-threaded acceleration. Finally, we report memory usage for both system and GPU, since they are important metrics when developing models for memory-constrained devices such as IoT devices and smartphones.

**Results**   We found that RETVec-raw and RETVec are the fastest vectorizers when a GPU is available and are in the top three when multi-core CPU are available (Table 1), which is common on current devices including smartphones. RETVec's performance on CPU stems from its ability to efficiently use the available cores, as visible in the Core Sec column – we expect that more optimized versions of SentencePiece and BPE could achieve similar performance.

# 5   Evaluation: Classification

In this section, we compare the classification training performance of RETVec against other vectorizers and word embeddings. We train the models from scratch and report classification accuracy on a wide range of datasets to benchmark the generality of the approach.

**Setup**   In order to perform a comprehensive evaluation, we evaluate classification performance on four different datasets with drastically different dataset sizes, number of languages, classification tasks, and text lengths, as summarized in Table 2. For example, the MASSIVE [8] intent classification dataset has very short sentences but a lot of languages (51), whereas the Yelp Reviews dataset has significantly longer texts but is monolingual.

We benchmark performance on three of the most common model architectures for text classification: Transformer (BERT-Mini [7]), CNN (DPCNN [15] ) and LSTM (Stacked-LSTM [1]), details provided in Appendix B. To keep evaluation fair and consistent when comparing various vectorizers, we keep the models the same and only swap the vectorizers. All models are implemented in TensorFlow 2.11 and training is conducted on a Google Cloud VM using a single NVidia V100 GPU.

All models are trained with Adam optimizer with $\beta_1 = 0.9$, $\beta_2 = 0.999$ and a max learning rate of 5e-4, with the exception of fastText on multilingual datasets. We found that a max learning rate of 5e-4 was too high and led to divergence, so we used 1e-4 instead for fastText on MASSIVE and Multilingual Amazon Reviews. For BERT, we linearly warmup the learning rate for 5k steps. All

models are trained for 100k steps with batch size 256 and cosine learning rate decay to 0. The test accuracy of each model averaged over 3 trials with different random seeds is reported in Table 3.

| Dataset | Model | Whitespace | SentencePiece | BPE | fastText | RETVec-raw | RETVec |
|---|---|---|---|---|---|---|---|
| AG News | RNN | 91.6% | 91.3% | 91.3% | **92.7%** | 91.3% | 92.6% |
| | CNN | **92.5%** | 92.4% | 90.7% | 92.3% | 88.2% | 91.2% |
| | BERT | 91.5% | 91.5% | 92.1% | 93.4% | 93.1% | **93.5%** |
| | AVG | 91.9% | 91.7% | 91.4% | **92.8%** | 90.9% | 92.4% |
| Yelp P. | RNN | 93.9% | 93.3% | 93.4% | 94.7% | 94.1% | **94.7%** |
| | CNN | **94.4%** | 93.8% | 93.0% | 93.9% | 92.5% | 93.4% |
| | BERT | 91.6% | 91.3% | 91.6% | **93.5%** | 93.2% | 92.8% |
| | AVG | 93.3% | 92.8% | 92.7% | **94.0%** | 93.3% | 93.6% |
| Multilingual Amazon P. | RNN | 93.2% | 90.7% | 90.3% | 87.0% | 92.9% | **93.5%** |
| | CNN | **93.2%** | 90.8% | 89.2% | 84.4% | 91.1% | 92.2% |
| | BERT | 91.8% | 87.3% | 87.2% | 86.6% | 92.7% | **92.6%** |
| | AVG | 92.8% | 89.6% | 88.9% | 86.0% | 92.2% | **92.8%** |
| MASSIVE | RNN | 70.3% | 69.6% | 68.6% | 13.5% | 71.0% | **73.8%** |
| | CNN | **70.4%** | 69.2% | 59.2% | 12.6% | 59.8% | 66.7% |
| | BERT | 70.6% | 67.8% | 66.2% | 23.9% | 78.1% | **78.6%** |
| | AVG | 70.4% | 68.9% | 64.7% | 16.7% | 69.6% | **73.0%** |
| Average | RNN | 87.3% | 86.2% | 85.9% | 72.0% | 87.3% | **88.7%** |
| | CNN | **87.6%** | 86.5% | 83.0% | 70.8% | 82.9% | 85.9% |
| | BERT | 86.4% | 84.5% | 84.3% | 74.4% | 89.3% | **89.4%** |
| | AVG | 87.1% | 85.8% | 84.4% | 72.4% | 86.5% | **88.0%** |

Table 3: Detailed classification results comparing different vectorizers when used to train models from scratch. **Bold** indicates best results, underline indicates second best.

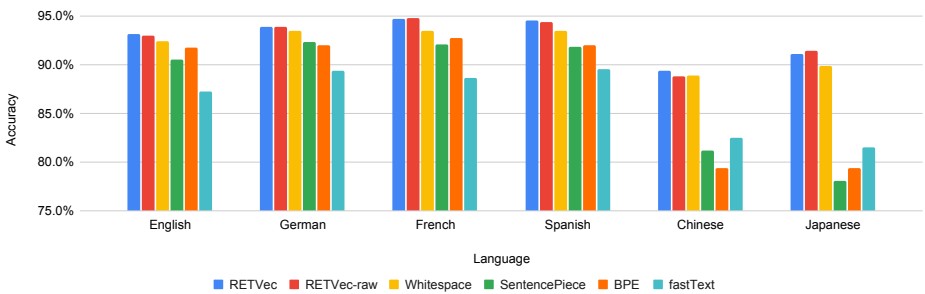

Figure 3: Classification performance on Multilingual Amazon Reviews broken down by language.

**Results** Overall, averaging over all datasets and model architectures, RETVec is the best performing vectorizer by some margin (0.9%). We observe that RETVec performs best when paired with Transformer and RNN architectures, but lags slightly behind for the CNN. Compared to RETVec-raw, RETVec's pre-trained model offers a significant performance improvement across all datasets and model architectures.

fastText performs the best on average for the English-only datasets, but does not perform well on the two multilingual datasets (especially MASSIVE), which explains its overall poor performance. Given that we are using the same code and models for all datasets and the fact that fastText results on English-only datasets match the ones reported in the original paper [16], we believe that fastText's poor performance on multilingual datasets is due to the fact that fastText vectors from different languages may not be compatible with each other when used as inputs to a multilingual model. However, we didn't find any mention of such an issue in the literature.

Delving deeper into multilingual capabilities, as plotted in Figure 3, we observe that RETVec consistently outperforms other vectorizers regardless of the language. We also observe that RETVec has a wider lead on non-Latin languages, which we attribute at least partially to our novel character encoder, given that RETVec and RETVec-raw achieve similar performance on them.

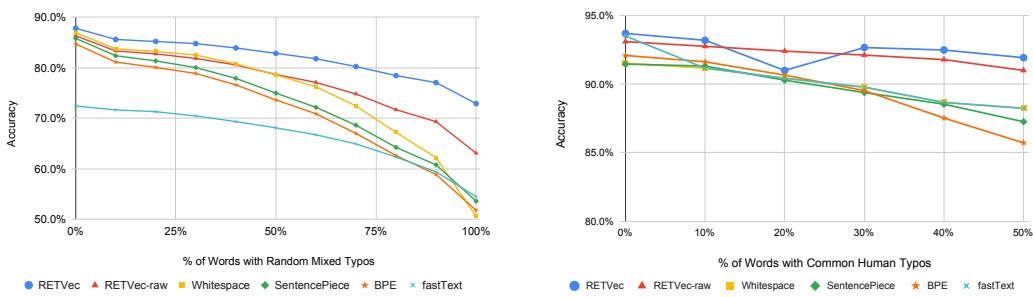

Figure 4: Comparison of various tokenizers' resilience against mixed random typos (left) and common human typos (right) when training classification models from scratch.

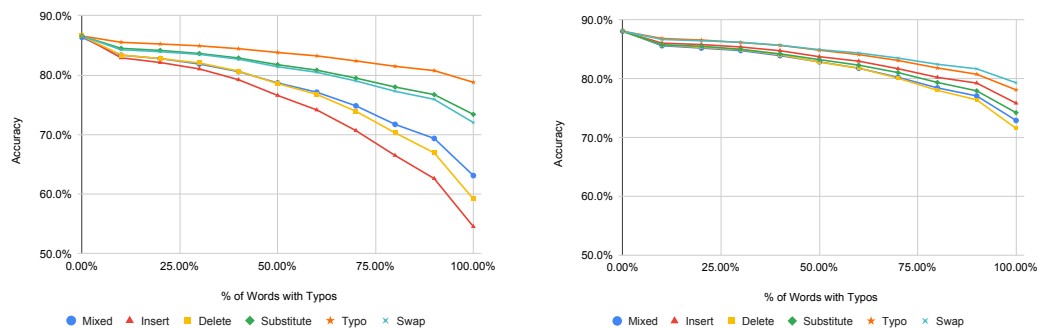

Figure 5: Comparison of RETVec resilience against various types of typos. RETVec-raw on the left, RETVec on the right.

## 6 Evaluation: Typo Resilience

In this section, we evaluate the resilience of the various vectorizers against both random typos and common human typos.

**Setup** We rely on two types of typo injection methods to evaluate the tokenizers' typo resilience: `random typos` and `common human typos`. For random typos, evaluation is performed on every model and dataset used in Section 5 and average accuracy on the typo-augmented test splits is recorded. For common human typos, because we only have them for English, we take the average accuracy across all three models on only AG News. We evaluate how each tokenizer's resilience degrades as the number of typos increases by gradually increasing the percentage of words with typos in the test set by increments of 10%.

Random typos are created by using a combination of insertion, deletion, substitution, neighboring swap, and keyboard-based typos. Each typo is applied with a block size of 1 or 2 characters. Words are selected at random and each word can only have one typo. To create human-like typos, we use the word replacement `noiser` from the Neuspell [14] package. The noiser uses 109k common misspellings of 17k popular English words. On the AG News test set, it can only inject typos into 55% of the words, so we limit the evaluation to 50% words with common human typos.

**Results** As reported in Figure 4, the performance of RETVec and RETVec-raw decreases significantly slower than the other vectorizers as the amount of typos increase. RETVec's pre-trained model increases RETVec resilience by up to 15% compared to RETVec-raw, demonstrating the effectiveness of RETVec's pre-training technique of using pair-wise metric learning to create syntactically robust word embeddings. SentencePiece and BPE perform about the same, with a gradual decline which keeps them competitive with RETVec-raw until about 60% random typo rate. We note that fastText and Whitespace provide more typo resilience than SentencePiece and BPE, which is expected because it has been previously noted that word-level vectorizers are typically more robust than character-level

| Tokenizer | Original Acc | Acc under Atk | Atk Success % |
|---|---|---|---|
| Whitespace | 89.4% | 32.3% | 63.9% |
| SentencePiece | 89.8% | 25.8% | 71.3% |
| BPE | 90.8% | 29.2% | 67.8% |
| fastText | 92.6% | 40.5% | 56.3% |
| RETVec-raw | 93.0% | 50.6% | 45.6% |
| RETVec | **93.7%** | **51.7%** | **44.8%** |

Table 4: Adversarial resilience evaluation for BERT-Mini trained on AG News with different vectorizers. Adversarial attack results are averaged across three types of character-level adversarial attacks: TextBugger [18], DeepWordBug [9], and Pruthi [23]. **Bold** indicates best results, underline indicates second best.

or subword-level vectorizers [23]. We see similar trends on the common human typo results, with RETVec-raw and RETVec being noticeably more resilient than baseline vectorizers when the typo rate is >30%.

Delving deeper into RETVec's typo resilience capabilities, as reported in Figure 5, character deletion is the hardest form of typo for RETVec to deal with. Without the pre-trained model, RETVec-raw also struggles with character insertion and RETVec's model also greatly helps nullify the impact of character swapping.

# 7 Evaluation: Adversarial Attack Resilience

**Setup**   We use the TextAttack framework [21] to evaluate the resilience of RETVec and other vectorizers against character-level adversarial attacks, specifically: TextBugger [18], DeepWordBug [9], and Pruthi [23]. For each model and vectorizer, the attacks are carried out on the same 1000 examples drawn randomly from the AG News test dataset.

**Results**   Table 4 demonstrates that RETVec models are significantly more resilient to adversarial text attacks than other embedding schemes. In particular, the average attack success rate against RETVec-raw and RETVec models are 45.6% and 44.8% respectively, which is significantly lower than for SentencePiece (71.3%) and BPE (67.8%). The best performing, non-RETVec, vectorizer is fastText (56.3% attack success rate), which makes sense because fastText handles OOV tokens by averaging the vector representations of the token's n-grams. Detailed results broken down by adversarial attack algorithm are reported in Appendix G.

# 8 Evaluation: Pre-training BERT

In this section, we evaluate RETVec's performance when used to pre-train transformers. We evaluate both RETVec's competitiveness on the GLUE benchmark [29] and typo resilience.

**Setup**   We use BERT-Base [7] for all experiments. Given that RETVec outputs word-level embeddings and is not restricted by a vocabulary size, we cannot directly use the standard BERT masked-language modeling (MLM) task and predict token ids using a softmax layer. Instead, we use an approximate MLM task for RETVec-based models by only masking and predicting input tokens that are in the top 100k words. Other than this, we follow the standard MLM procedure by selecting 15% of tokens at random and replacing a selected token with (1) the [MASK] token with 80% chance (2) a random token with 10% chance (3) the same token unchanged with 10% chance. For the baseline, we use SentencePiece with a vocabulary size of 32k and the standard MLM task described in [7].

For each model, we pre-train for 1M steps with batch size 64 on the English C4 dataset [32]. We use Adam with a max learning rate of 5e-5, $\beta_1 = 0.9$, $\beta_2 = 0.999$, and 0.01 L2 weight decay. We warmup for 10000 steps and linearly decay the learning rate. We fine-tune all models for 20 epochs on GLUE using 3 different random seeds and report the best result for each dataset. We pre-train and fine-tune each model using 8 NVidia V100s. Detailed pre-training and fine-tuning hyperparameters can be found in Appendix H.

To benchmark typo resilience of RETVec-based pre-trained BERT models, we fine-tune on AG News to produce comparable results to Section 6, and follow the same evaluation methodology against random and common human typos.

| Vectorizer | MNLI | QNLI | QQP | RTE | SST-2 | MRPC | CoLA | STS-B | Avg |
|---|---|---|---|---|---|---|---|---|---|
| SentencePiece | 80.6 | 88.7 | 90.1 | **67.2** | 91.1 | 85.8 | **50.7** | **82.5** | **79.6** |
| RETVec-raw | **82.5** | **89.6** | **90.5** | 65.3 | **91.5** | 87.2 | 49.1 | 79.3 | 79.4 |
| RETVec | 81.4 | 89.2 | 90.4 | 65.7 | 90.8 | **87.3** | 47.9 | 79.8 | 79.1 |

Table 5: Results on GLUE dev sets. We report matched accuracy for MNLI, Matthews correlation for CoLA, Pearson correlation for STS-B, F1 score for QQP, and accuracy for all other tasks. **Bold** indicates best results, underline indicates second best.

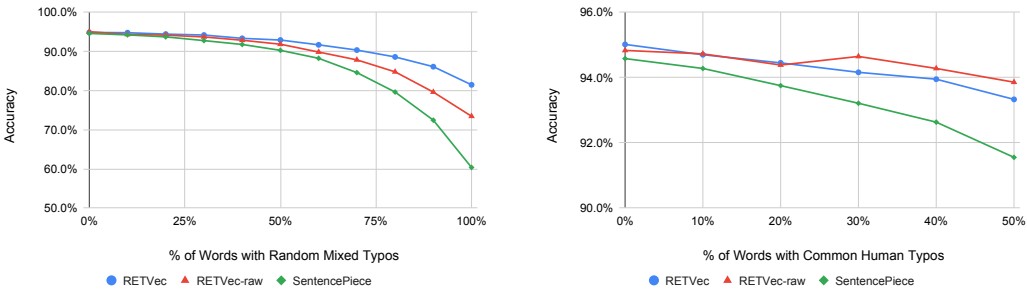

Figure 6: Comparison of typo resilience against mixed random typos (left) and common human typos (right) when fine-tuning a pre-trained BERT-Base model on AG News dataset for RETVec and SentencePiece.

**Results**  Overall, as reported in Table 5, BERT pre-trained with RETVec and RETVec-raw using MLM on 100k words achieves competitive results compared to SentencePiece. Additionally, using RETVec reduces the pre-training time of BERT by around 10% in our experiment, when training for an equivalent number of steps. Surprisingly, RETVec-raw is more competitive than RETVec, which we attribute to the fact that BERT-Base has more than enough capacity to learn a competitive representation without RETVec's word embedding model. However, this increased competitiveness for RETVec-raw comes at the cost of decreased typo resilience, as reported in Figure 6, although the difference in typo resilience between pre-trained BERT models using RETVec-raw versus RETVec is only significant when the percentage of injected typos exceeds 50% of the words. As such, it seems that using RETVec-raw when pre-training large models is the best choice. How to optimally do so is left for future work.

## 9  Evaluation: In the Wild

As a final evaluation, to ensure RETVec will work in practice against adversarial content, we trained a version of a transformer model we use as part of our spam email filtering system at Google. As with previous experiments, we kept everything constant besides swapping SentencePiece with RETVec. Benchmarking on our internal production evaluation dataset, we found that RETVec improved the recall at 0.80% false positive rate by ∼0.51%, helping detect 38% of the spam emails that would have been missed by the baseline model while reducing model serving latency by 30%. This result gives us confidence that RETVec is competitive for real-world classification tasks, especially in the face of adversarial content.

## 10  Ablation Study

In this section, we report how key hyperparameters variations affect RETVec's performance. An unforeseen challenge we discovered while developing RETVec models is that a lower pre-training loss didn't directly translate into meaningful performance on our benchmarks. To overcome this

challenge, we had to include an extra validation step that trains and evaluates models on the Multilingual Amazon Reviews dataset [19] using the methodology described in Section 5 to measure downstream improvement. We choose the Amazon dataset due to its multilingual nature and difficulty. Detailed results on classification performance by model type, pre-training loss, and model size for the ablation studies can be found in Appendix C, along with additional experiments on less impactful hyperparameters.

| Architecture | Loss | Accuracy | | Emb Dim | Loss | Acc | | Word Len | Loss | Acc |
|---|---|---|---|---|---|---|---|---|---|---|
| **MLP** | 0.0248 | 92.89% | | 100 | 0.0267 | 92.6% | | 12 | 0.0267 | 92.8% |
| BERT + MLP | 0.0129 | 92.54% | | 128 | 0.0260 | 92.7% | | 14 | 0.0259 | 92.8% |
| T5 + MLP | 0.0120 | 92.72% | | 200 | 0.0253 | 92.7% | | **16** | 0.0248 | 92.9% |
| GAU + MLP | 0.0133 | 92.82% | | **256** | 0.0248 | 92.9% | | 20 | 0.0242 | 92.7% |
| LSTM + MLP | 0.0179 | 92.61% | | 300 | 0.0245 | 92.8% | | 24 | 0.0234 | 92.9% |
| CNN + MLP | 0.0214 | 92.72% | | 384 | 0.0237 | 92.8% | | 32 | 0.0235 | 92.8% |

Table 6: Ablation studies for RETVec architecture (left), word embedding dimension (middle), and word length (right). **Bold** denotes the hyperparameter selected for the final RETVec model.

**Model architecture** Despite considerable efforts, we couldn't find a more complex architecture that outperformed the simple MLP architecture (Table 6). The best transformer-based architecture which uses 2 encoder blocks of dimension 128 resulted in significantly lower loss regardless of the transformer block used (BERT [7], GAU [13], T5 [32]). However, as alluded to earlier, the significantly lower loss did not translate to better performance on the benchmarks. We hypothesize that the fitting the text manifold is an easy enough task to solve with any reasonable architecture.

**Embedding Size** We train RETVec with different embedding dimensions and report the results in Table 6. Classification benchmarks show little difference in performance for embedding dimensions between 128 and 512, with 256 embedding dimension having the slightest edge.

**Max Input Character length** To select the optimal max word length for RETVec, we started by looking the fastText dataset's word length distribution. More than 95% of the words are less than 16 characters long and the median word length is 7.9 characters (Appendix I). Using more than 16 characters didn't led to any accuracy improvements despite those longer inputs exhibiting a lower loss, as reported in Table 6.

## 11 Discussion and Future Work

In this paper, we presented RETVec, a new multilingual text vectorizer that combines a novel character encoding with an small model to project words into a compact, 256-dimensional embedding. Through extensive evaluations, we demonstrated that models trained using RETVec achieve state-of-the-art classification performance and provide a 10-15% increase in resilience against typos and adversarial attacks compared to popular text vectorizers and word embeddings.

The key outstanding question is how to best utilize RETVec for generative tasks and in large language models (LLMs), which could enable us to train LLMs with stronger multilingual capabilities, improved adversarial robustness, and reduced model size and computational costs. In particular, for "small" LLMs (less than or around 1 billion total parameters), the vocabulary embedding layer can be more than 20% of the total parameters [3], which would be eliminated when using RETVec.

We discovered that the main difficulty when training generative models using RETVec is that RETVec's 256-float embedding cannot be directly converted into a softmax output, unlike other tokenizers which represent text as integer token IDs. In order to effectively use RETVec in decoder-only models, we expect that a new training procedure for RETVec-based models which is compatible with text generation tasks needs to be invented, e.g. a pre-training procedure which does not rely on directly predicting the next token. We briefly experimented with ideas including decoding the RETVec representation character-by-character or using a VQ-VAE [28] model to quantize the output, but the results so far were inconclusive. We hope to explore this area and address this limitation in future work. Other potential directions of future work include using RETVec as a word embedding in various applications in place of GloVe [22] and word2vec [20], and using the RETVec character encoder and training procedure to train text similarity models.

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

# Appendix

## A    RETVec Model Details

Table 7 details the hyperparameter settings for the RETVec model architecture, as described in Section 3.

| Hyperparameter | RETVec |
|---|---|
| Max word length | 16 |
| Per-character encoding dim | 24 |
| Activation | GeLU |
| # of projection layers | 1 |
| Projection layer dim | 32 |
| # of fully-connected layers | 2 |
| Fully-connected layer dim | 256 |
| Spatial dropout rate | 0.0625 |
| Dropout rate | 0 |
| Embedding activation | Tanh |
| Embedding dim | 256 |
| Similarity dim | 256 |

Table 7: RETVec model hyperparameter details.

## B    Benchmarking Models

In this section, we provide more detailed model hyperparameters for the evaluation models used in Section 5 and Section 8.

**RNN**    We used a Stacked-LSTM architecture, with the following hyperparameters:

- Dim: 256
- Layers: 4
- Dropout rate: 0.1

**DPCNN**    We use the architecture described in [15], with the following hyperparameters:

- Filters: 256
- Layers: 6
- Kernel size: 3
- Final dropout: 0.5
- Activation: ReLU

**BERT-Mini**    We use the architecture as described in [7] for BERT-Mini, with the following hyperparameters:

- Layers: 4
- Hidden dim: 256
- Intermediate dim: 1024
- Self-attention heads: 4
- Dropout rate: 0.1
- Activation: GeLU

**BERT-Base** We use the architecture described in [7] for BERT-Base, with the following hyperparameters:

- Layers: 12
- Hidden dim: 768
- Intermediate dim: 3072
- Self-attention heads: 12
- Dropout rate: 0.1
- Activation: GeLU

## C RETVec Ablation Studies

In this section, we present our ablation study results for the RETVec model design and hyperparameter selection. Results are reported on the Multilingual Amazon Reviews dataset following the methodology described in Section 10.

### C.1 Embedding Dimension

Detailed results on how RETVec's embedding layer dimension affects classification performance are reported in Table 8.

| Embedding Dim | Pre-training Loss | # Params | Test Accuracy | | | |
|---|---|---|---|---|---|---|
| | | | RNN | CNN | BERT | AVG |
| 64 | 0.0284 | 181k | 92.8% | 91.8% | 92.5% | 92.4% |
| 100 | 0.0267 | 190k | 93.2% | 92.1% | 92.4% | 92.6% |
| 128 | 0.0260 | 197k | 93.3% | 92.2% | 92.5% | 92.7% |
| 200 | 0.0253 | 216k | 93.4% | 92.3% | 92.6% | 92.7% |
| **256** | 0.0248 | 230k | 93.6% | 92.3% | 92.8% | 92.9% |
| 300 | 0.0245 | 241k | 93.6% | 92.3% | 92.6% | 92.8% |
| 384 | 0.0237 | 263k | 93.6% | 92.2% | 92.6% | 92.8% |
| 512 | 0.0225 | 296k | 93.7% | 92.3% | 92.8% | 92.9% |

Table 8: Ablation study results on the effect of the RETVec pre-trained model's embedding dimension on classification performance. **Bold** denotes the hyperparameter selected for the final RETVec model.

### C.2 Model Architecture

Detailed results on the effect of RETVec architecture type on classification performance are reported in Table 9.

| Architecture | Pre-training Loss | Test Accuracy | | | |
|---|---|---|---|---|---|
| | | RNN | CNN | BERT | AVG |
| **MLP** | 0.0248 | 93.6% | 92.3% | 92.8% | 92.9% |
| MLP + BERT | 0.0129 | 93.2% | 92.0% | 92.4% | 92.5% |
| MLP + T5 | 0.0120 | 93.5% | 92.2% | 92.5% | 92.7% |
| MLP + GAU | 0.0133 | 93.5% | 92.2% | 92.8% | 92.8% |
| MLP + LSTM | 0.0179 | 93.4% | 91.9% | 92.5% | 92.6% |
| MLP + CNN | 0.0214 | 93.5% | 92.1% | 92.6% | 92.7% |

Table 9: Ablation study results on RETVec model architecture type on classification performance. **Bold** denotes the hyperparameter selected for the final RETVec model.

### C.3 Maximum Word length

Detailed results on how RETVec's maximum input word length affects classification performance are reported in Table 10.

| Word Len | Pre-training Loss | # Params | Test Accuracy | | | |
|---|---|---|---|---|---|---|
| | | | RNN | CNN | BERT | AVG |
| 8 | 0.0428 | 181k | 93.6% | 92.3% | 92.7% | 92.8% |
| 10 | 0.0326 | 193k | 93.5% | 92.4% | 92.6% | 92.8% |
| 12 | 0.0267 | 206k | 93.5% | 92.2% | 92.7% | 92.8% |
| 14 | 0.0259 | 218k | 93.5% | 92.2% | 92.6% | 92.8% |
| **16** | 0.0248 | 230k | 93.6% | 92.3% | 92.8% | 92.9% |
| 20 | 0.0242 | 255k | 93.5% | 92.2% | 92.6% | 92.7% |
| 24 | 0.0234 | 280k | 93.6% | 92.3% | 92.7% | 92.9% |
| 28 | 0.0246 | 304k | 93.5% | 92.2% | 92.7% | 92.8% |
| 32 | 0.0235 | 328k | 93.6% | 92.2% | 92.6% | 92.8% |

Table 10: Ablation study results on the effect of maximum word length on RETVec classification performance. **Bold** denotes the hyperparameter selected for the final RETVec model.

## C.4 Pre-training Loss Hyperparameters

Detailed results on the effect of Multi-Similarity loss hyperparameters on RETVec classification performance are reported in Table 11. We also experimented with Circle Loss [27] and report the results in Table 12.

| Hyperparameter | | | | Test Accuracy | | | |
|---|---|---|---|---|---|---|---|
| $\alpha$ | $\beta$ | $\lambda$ | Pre-training Loss | RNN | CNN | BERT | AVG |
| 2 | 20 | 0.5 | 0.0432 | 93.5% | 92.1% | 92.4% | 92.7% |
| 2 | 20 | 1.0 | 0.2643 | 92.9% | 91.2% | 92.6% | 92.2% |
| 2 | 40 | 0.5 | 0.0455 | 93.6% | 92.1% | 92.6% | 92.7% |
| 2 | 40 | 1.0 | 0.3610 | 92.9% | 91.2% | 92.4% | 92.2% |
| 2 | 80 | 0.5 | 0.0464 | 93.5% | 92.3% | 92.6% | 92.8% |
| 2 | 80 | 1.0 | 0.3583 | 92.6% | 91.1% | 92.4% | 92.0% |
| 4 | 20 | 0.5 | 0.0270 | 93.4% | 92.0% | 92.6% | 92.7% |
| 4 | 20 | 1.0 | 0.1537 | 93.0% | 91.3% | 92.4% | 92.2% |
| 4 | 40 | 0.5 | 0.0242 | 93.5% | 92.4% | 92.6% | 92.8% |
| 4 | 40 | 1.0 | 0.1919 | 92.8% | 91.3% | 92.4% | 92.2% |
| **4** | **80** | **0.5** | 0.0248 | 93.6% | 92.3% | 92.8% | 92.9% |
| 4 | 80 | 1.0 | 0.1851 | 92.8% | 91.2% | 92.4% | 92.1% |

Table 11: Ablation study results on the effect of Multi-Similarity loss hyperparameters on RETVec classification performance. $\epsilon = 0.1$ is fixed for all experiments. **Bold** denote the hyperparameters selected for the final RETVec model.

| Circle-Loss Hyperparameter | | | Test Accuracy | | | |
|---|---|---|---|---|---|---|
| Scale Factor $\gamma$ | Relaxation Factor $m$ | Pre-training Loss | RNN | CNN | BERT | AVG |
| 64 | 0.3 | 7.55 | 93.3% | 91.9% | 92.6% | 92.6% |
| 64 | 0.4 | 2.77 | 93.5% | 92.0% | 92.4% | 92.7% |
| 64 | 0.5 | 0.63 | 93.6% | 92.4% | 92.6% | 92.9% |
| 128 | 0.3 | 12.85 | 93.3% | 91.9% | 92.6% | 92.6% |
| 128 | 0.4 | 5.06 | 93.6% | 92.1% | 92.6% | 92.8% |
| 128 | 0.5 | 1.18 | 93.6% | 92.3% | 92.6% | 92.9% |
| 256 | 0.3 | 24.97 | 93.3% | 91.9% | 92.5% | 92.5% |
| 256 | 0.4 | 9.49 | 93.5% | 92.0% | 92.4% | 92.7% |
| 256 | 0.5 | 2.59 | 93.7% | 92.4% | 92.5% | 92.9% |
| 512 | 0.3 | 49.01 | 93.2% | 92.0% | 92.6% | 92.6% |
| 512 | 0.4 | 19.88 | 93.3% | 92.2% | 92.7% | 92.7% |
| 512 | 0.5 | 5.26 | 93.6% | 92.4% | 92.7% | 92.9% |

Table 12: Ablation study results for pre-training RETVec with various Circle Loss [27] hyperparameter settings.

## C.5 Model Capacity

Detailed results on how the number and dimension of fully-connected dense layers in the RETVec model affects classification performance are presented in Table 13.

| Model Capacity | | # Params | Pre-training Loss | Test Accuracy | | | |
|---|---|---|---|---|---|---|---|
| Dense Layers | Dense Layer Dim | | | RNN | CNN | BERT | AVG |
| 0 | - | 99k | 0.0445 | 92.8% | 91.6% | 92.3% | 92.2% |
| 1 | 128 | 82k | 0.0383 | 93.5% | 91.8% | 92.4% | 92.6% |
| 1 | 256 | 164k | 0.0312 | 93.5% | 91.8% | 92.4% | 92.6% |
| 1 | 384 | 246k | 0.0284 | 93.6% | 92.0% | 92.6% | 92.7% |
| 1 | 512 | 328k | 0.0258 | 93.6% | 92.2% | 92.8% | 92.8% |
| 2 | 128 | 989k | 0.0334 | 93.4% | 91.9% | 92.6% | 92.6% |
| **2** | **256** | 230k | 0.0248 | 93.6% | 92.3% | 92.8% | 92.9% |
| 2 | 384 | 394k | 0.0201 | 93.5% | 92.3% | 92.6% | 92.8% |
| 2 | 512 | 591k | 0.0177 | 93.7% | 92.3% | 92.6% | 92.9% |
| 3 | 128 | 115k | 0.0314 | 93.4% | 91.9% | 92.5% | 92.6% |
| 3 | 256 | 296k | 0.0213 | 93.6% | 92.4% | 92.5% | 92.8% |
| 3 | 384 | 542k | 0.0175 | 93.6% | 92.2% | 92.7% | 92.8% |
| 3 | 512 | 854k | 0.0149 | 93.5% | 92.4% | 92.6% | 92.8% |

Table 13: Ablation study results on the effect of RETVec model capacity (number and dimension of the fully-connected layers) on classification performance. **Bold** denote the hyperparameters selected for the final RETVec model.

## C.6 Spatial Dropout Rate

Detailed ablation study results on the amount of spatial dropout in the RETVec model and its effect on classification performance are presented in Table 14. Increments of $1/16$ were used because it corresponds to dropping out one character of the input on average, since RETVec's model accepts an input of up to 16 characters per word.

| Spatial Dropout | Pre-training Loss | Test Accuracy | | | |
|---|---|---|---|---|---|
| | | RNN | CNN | BERT | AVG |
| 0.00% | 0.0122 | 93.4% | 91.4% | 92.5% | 92.4% |
| **6.25%** | 0.0248 | 93.6% | 92.3% | 92.8% | 92.9% |
| 12.50% | 0.0465 | 93.4% | 92.0% | 92.3% | 92.6% |
| 18.75% | 0.0722 | 92.7% | 91.5% | 91.8% | 92.0% |
| 25.00% | 0.0967 | 92.7% | 91.4% | 91.2% | 91.8% |

Table 14: Ablation study results on the effect of spatial dropout rate on the RETVec input character encoding. **Bold** denotes the value selected for the final RETVec model.

## C.7 Pre-Training Objectives

We evaluated combining RETVec's pre-training objective (Multi-Similarity loss) with other objective functions and pre-training tasks as well. Specifically, we experimented with the following objectives: 1) augmentation position prediction, 2) augmentation position and type prediction, 3) decoding (predicting the character encoding of the input token), and 4) denoising (predicting the character encoding of the original, non-augmented token). Table 15 reports the results of our experiments on different pre-training objectives.

| Objectives | Similarity Loss | Total Loss | RNN | CNN | BERT | AVG |
|---|---|---|---|---|---|---|
| Similarity, Augmentation Position Detection | 0.0261 | 0.2228 | 93.4% | 92.0% | 92.8% | 92.7% |
| Similarity, Augmentation Position and Type Prediction | 0.0238 | 0.0970 | 93.2% | 92.2% | 92.6% | 92.7% |
| Similarity, Decoding | 0.0278 | 0.0384 | 93.5% | 92.1% | 92.6% | 92.8% |
| Similarity, Denoising | 0.0241 | 0.1088 | 93.3% | 91.8% | 92.7% | 92.6% |
| Similarity | 0.0248 | 0.0248 | 93.6% | 92.3% | 92.8% | 92.9% |

Table 15: Ablation study results on combining different pre-training objectives with similarity loss for RETVec pre-training.

# D    RETVec Pre-training Dataset Augmentations

Below, we provide the full list of character-level augmentations (broken down into four categories) used to generate typo-augmented words for the RETVec pre-training dataset, as described in Section 3.3.

- Deletion
- Insertion
    - Repeated character insertion
    - n-grams based prefix insertion for $n = 3, 4, 5$
    - n-grams based suffix insertion for $n = 3, 4, 5$
    - Random ASCII character insertion
    - Language alphabet-based random character insertion
    - Random punctuation insertion
    - Random punctuation prefix
    - Random punctuation suffix
    - Random BMP Unicode insertion
    - Random emoji prefix
    - Random emoji suffix
- Substitution
    - Case substitution
    - n-grams based substitution for $n = 3, 4, 5$
    - QWERTY keyboard typo substitution
    - Homoglyphs substition
    - Random ASCII character substitution
    - Language alphabet-based random character subsitution
    - Random puctuation substitution
    - Random BMP Unicode substitution
- Transposition
    - Neighboring character swap
    - 3-character block random shuffle

# E    RETVec Pre-training Hyperparameters

We train RETVec using Multi-Similarity loss with hyperparameters $\alpha = 4$, $\beta = 40$, $\epsilon = 0.1$ and $\lambda = 0.5$. Detailed pre-training hyperparameters are reported in Table 16.

| Hyperparameter | Pre-training |
|---|---|
| Training steps | 500k |
| Batch size | 1024 |
| Adam $\epsilon$ | 1.00e-7 |
| Adam $\beta_1$ | 0.9 |
| Adam $\beta_2$ | 0.999 |
| Weight decay | 0 |
| Peak learning rate | 0.001 |
| End learning rate | 0.0001 |
| Warmup steps | 10000 |
| Decay function | Cosine |

Table 16: RETVec pre-training optimizer hyperparameters.

| Dataset | Vectorizer | 0% | 10% | 20% | 30% | 40% | 50% | 60% | 70% | 80% | 90% | 100% |
|---|---|---|---|---|---|---|---|---|---|---|---|---|
| AG News | Whitespace | 92.0% | 91.6% | 91.0% | 90.2% | 88.6% | 86.8% | 83.8% | 79.3% | 72.3% | 63.5% | 49.1% |
| | SentencePiece | 91.8% | 91.0% | 89.9% | 88.5% | 86.2% | 82.8% | 78.7% | 74.0% | 67.9% | 60.9% | 51.5% |
| | BPE | 91.6% | 90.5% | 89.1% | 87.6% | 84.8% | 80.6% | 76.1% | 70.2% | 63.8% | 56.2% | 46.3% |
| | fastText | 92.8% | 92.1% | 91.6% | 90.7% | 89.1% | 87.8% | 85.1% | 82.5% | 77.2% | 70.5% | 59.7% |
| | RETVec-raw | 91.0% | 90.3% | 89.6% | 88.3% | 87.3% | 85.5% | 83.7% | 81.8% | 78.2% | 74.7% | 68.5% |
| | RETVec | 92.6% | 92.1% | 91.6% | 91.1% | 90.5% | 89.9% | 88.7% | 87.3% | 86.0% | 84.2% | 81.2% |
| Yelp P. | Whitespace | 93.2% | 92.5% | 91.6% | 90.5% | 88.6% | 86.7% | 84.1% | 80.4% | 75.8% | 69.7% | 60.9% |
| | SentencePiece | 93.1% | 91.5% | 89.7% | 87.7% | 85.1% | 81.9% | 78.7% | 75.1% | 71.2% | 67.4% | 62.6% |
| | BPE | 93.0% | 91.7% | 90.2% | 88.5% | 86.2% | 83.5% | 80.5% | 77.2% | 73.1% | 69.0% | 64.2% |
| | fastText | 94.1% | 93.5% | 92.8% | 92.0% | 90.7% | 89.3% | 87.9% | 85.8% | 83.1% | 80.2% | 75.6% |
| | RETVec-raw | 92.4% | 91.7% | 90.9% | 89.8% | 88.6% | 87.2% | 85.7% | 83.7% | 81.2% | 78.6% | 74.5% |
| | RETVec | 92.7% | 92.2% | 91.6% | 90.9% | 90.0% | 89.2% | 88.0% | 86.8% | 85.1% | 83.5% | 80.9% |
| Multilingual Amazon P. | Whitespace | 92.7% | 92.1% | 91.6% | 91.1% | 90.2% | 89.1% | 87.9% | 86.2% | 83.7% | 80.9% | 75.0% |
| | SentencePiece | 89.6% | 88.5% | 87.5% | 86.2% | 84.5% | 82.6% | 80.7% | 78.7% | 76.1% | 73.9% | 70.7% |
| | BPE | 88.9% | 87.7% | 86.6% | 85.4% | 83.6% | 81.8% | 80.1% | 77.9% | 75.3% | 73.4% | 70.3% |
| | fastText | 86.2% | 85.5% | 84.8% | 84.0% | 82.9% | 81.0% | 80.1% | 77.9% | 75.9% | 73.9% | 70.2% |
| | RETVec-raw | 92.3% | 91.6% | 90.8% | 90.0% | 89.0% | 87.7% | 86.3% | 84.8% | 82.7% | 80.6% | 77.0% |
| | RETVec | 92.9% | 92.5% | 92.0% | 91.7% | 91.3% | 90.6% | 90.2% | 89.5% | 88.6% | 87.7% | 86.1% |
| MASSIVE | Whitespace | 70.0% | 58.9% | 58.8% | 58.4% | 55.8% | 51.7% | 49.2% | 43.9% | 37.1% | 34.8% | 17.3% |
| | SentencePiece | 69.0% | 58.5% | 58.4% | 57.8% | 55.8% | 52.7% | 50.7% | 46.7% | 41.8% | 40.9% | 29.6% |
| | BPE | 65.5% | 54.6% | 54.4% | 54.1% | 52.0% | 48.6% | 46.9% | 42.7% | 38.4% | 37.0% | 26.4% |
| | fastText | 16.7% | 15.5% | 16.1% | 15.2% | 14.8% | 14.3% | 13.9% | 13.1% | 13.1% | 13.0% | 12.5% |
| | RETVec-raw | 69.6% | 59.7% | 59.7% | 59.2% | 57.4% | 54.4% | 52.7% | 49.0% | 44.8% | 43.6% | 32.5% |
| | RETVec | 73.2% | 65.7% | 65.7% | 65.5% | 63.9% | 61.8% | 60.4% | 57.5% | 54.1% | 52.9% | 43.5% |

Table 17: Random mixed typo resilience results (0% to 100% word typo rate) for each classification dataset and vectorizer. Following the methodology described in Section 6, test accuracy on each dataset is reported and results are averaged across the three model architectures we benchmarked in 5.

# F Typo Resilience Evaluation

Detailed results for random mixed typo resilience across every dataset and vectorizer can be found in Table 17.

# G Adversarial Resilience Evaluation

We report adversarial attack resilience results for all vectorizers, classification models, and adversarial attack algorithms we benchmarked in Table 18. The TextAttack [21] framework was used to conduct all three types of adversarial attacks.

| Model | Vectorizer | TextBugger | | | Pruthi | | | DeepWordBug | | |
|---|---|---|---|---|---|---|---|---|---|---|
| | | Original Acc | Acc under Atk | Atk Success % | Original Acc | Acc under Atk | Atk Success % | Original Acc | Acc under Atk | Atk Success % |
| LSTM | Whitespace | 90.6% | 9.9% | 89.1% | 90.6% | 84.2% | 7.1% | 90.6% | 9.9% | 89.1% |
| | SentencePiece | 90.3% | 0.8% | 99.1% | 90.3% | 68.1% | 24.6% | 90.3% | 0.8% | 99.1% |
| | BPE | 88.1% | 3.3% | 96.3% | 88.1% | 73.3% | 16.8% | 88.1% | 3.3% | 96.3% |
| | fastText | 92.7% | 14.4% | 84.5% | 92.7% | 83.3% | 10.1% | 92.7% | 14.4% | 84.5% |
| | RETVec-raw | 90.8% | 22.3% | 75.4% | 90.8% | 74.8% | 17.6% | 90.8% | 22.3% | 75.4% |
| | RETVec | 91.8% | 23.7% | 74.2% | 91.8% | 80.9% | 11.9% | 91.8% | 23.7% | 74.2% |
| CNN | Whitespace | 90.9% | 17.6% | 80.6% | 90.9% | 83.8% | 7.8% | 90.9% | 9.0% | 90.1% |
| | SentencePiece | 90.3% | 2.9% | 96.8% | 90.3% | 72.2% | 20.0% | 90.3% | 3.5% | 96.1% |
| | BPE | 89.3% | 31.8% | 64.4% | 89.3% | 54.1% | 39.4% | 89.3% | 43.5% | 51.3% |
| | fastText | 91.9% | 17.3% | 81.2% | 91.9% | 74.9% | 18.5% | 91.9% | 15.4% | 83.2% |
| | RETVec-raw | 86.9% | 30.1% | 65.4% | 86.9% | 59.6% | 31.4% | 86.9% | 37.7% | 56.6% |
| | RETVec | 91.4% | 34.3% | 62.5% | 91.4% | 77.4% | 15.3% | 91.4% | 45.0% | 50.8% |
| BERT | Whitespace | 89.4% | 9.8% | 89.0% | 89.4% | 83.6% | 6.5% | 89.4% | 3.3% | 96.3% |
| | SentencePiece | 89.8% | 2.8% | 96.9% | 89.8% | 70.8% | 21.2% | 89.8% | 3.7% | 95.9% |
| | BPE | 90.8% | 8.2% | 91.0% | 90.8% | 78.2% | 13.9% | 90.8% | 1.2% | 98.7% |
| | fastText | 92.6% | 22.9% | 75.3% | 92.6% | 80.5% | 13.1% | 92.6% | 18.1% | 80.5% |
| | RETVec-raw | 93.0% | 30.8% | 66.9% | 93.0% | 82.2% | 11.6% | 93.0% | 38.9% | 58.2% |
| | RETVec | 93.7% | 30.1% | 67.9% | 93.7% | 84.6% | 9.7% | 93.7% | 40.5% | 56.8% |

Table 18: Detailed adversarial resilience results on AG News. Results are reported on the same randomly selected 1000 examples from the AG News test split, following the methodology described in Section 7.

# H Pre-training and Fine-tuning BERT

Table 19 details the hyperparameter settings used for pre-training and fine-tuning BERT-Base models. Table 20 shows detailed results on the GLUE benchmark, including the models' average performance and standard deviation for each GLUE task.

| Hyperparameter | Pre-training | Fine-tuning |
|---|---|---|
| Training steps | 100k steps | 20 epochs |
| Batch size | 64 | 32 |
| Sequence length | 512 | 512 |
| Adam $\epsilon$ | 1e-8 | 1e-8 |
| Adam $\beta_1$ | 0.9 | 0.9 |
| Adam $\beta_2$ | 0.999 | 0.999 |
| Weight decay | 0.01 | 0.01 |
| Max learning rate | 5e-5 | 2e-5 |
| End learning rate | 0 | 0 |
| Warmup steps | 10000 | First 5% of steps |
| Decay function | Linear | None |

Table 19: Pre-training and fine-tuning hyperparameters for BERT-Base models described in Section 8.

| Vectorizer | MNLI | QNLI | QQP | RTE | SST-2 | MRPC | CoLA | STS-B | GLUE Avg |
|---|---|---|---|---|---|---|---|---|---|
| SentencePiece | 80.6 (0.1) | 88.4 (0.3) | 90.1 (0.0) | **66.1 (1.0)** | 90.8 (0.3) | 85.4 (0.15) | **50.3 (0.4)** | **82.0 (0.5)** | **79.2 (0.3)** |
| RETVec-raw | **82.0 (0.5)** | **89.5 (0.1)** | **90.4 (0.1)** | 64.5 (0.9) | **91.5 (0.1)** | 86.3 (0.9) | 47.9 (1.1) | 79.1 (0.2) | 78.9 (0.4) |
| RETVec | 80.9 (0.4) | 88.9 (0.3) | 90.4 (0.1) | 65.0 (0.7) | 90.7 (0.2) | **86.9 (0.4)** | 47.2 (0.7) | 79.6 (0.2) | 78.7 (0.3) |

Table 20: Detailed results on GLUE Benchmark for pre-trained BERT-Base models using RETVec compared to SentencePiece. Each model is trained three times with different seeds, and the average and standard deviation is reported here. **Bold** indicates best results, underline indicates second best.

# I  fastText Word Dataset

Table 21 contains statistics on word length computed on the fastText word dataset using words from all 157 available languages.

| Avg | Median | Std | p90 | p95 | p99 | p99.9 |
|---|---|---|---|---|---|---|
| 8.4 | 7.9 | 4.6 | 13.0 | **15.0** | 20.8 | **36.1** |

Table 21: Word length statistics computed on all fastText words from 157 languages. p90 denotes the 90th percentile, p95 denotes the 95th percentile, and so on.

# J  Embedding Visualization

We use the TensorBoard Embedding Projector to visualize RETVec embeddings for words, as shown in Figure J.

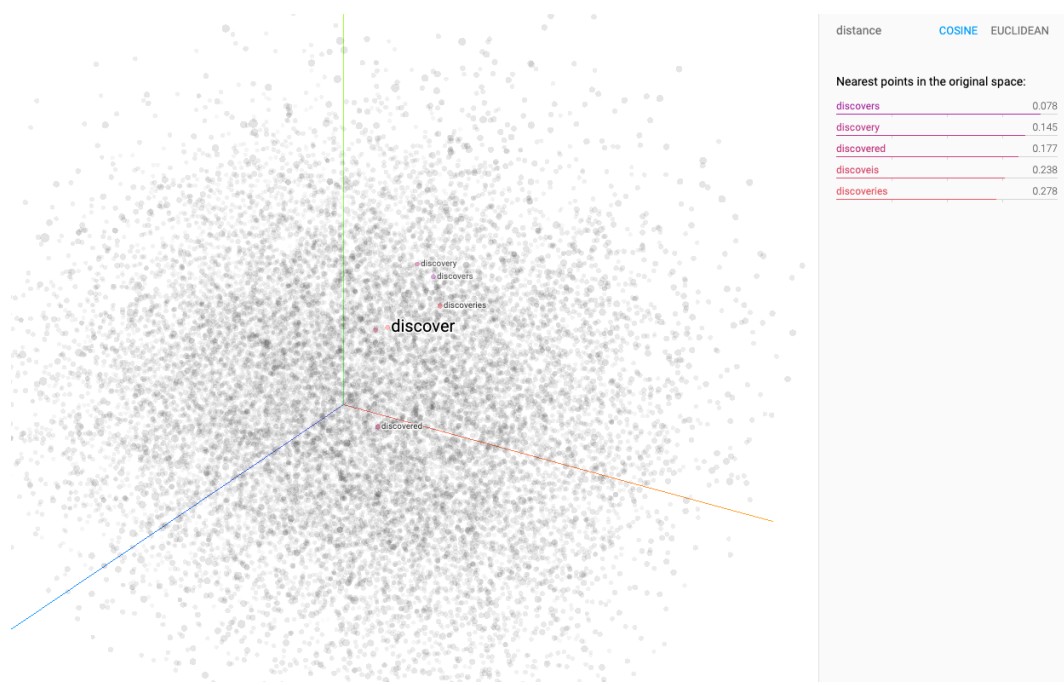

Figure 7: RETVec embedding visualization (using the TensorBoard Embedding Projector) of the most common 10000 English words and a typo-laden version of each word. The word 'discover' is selected as an example, and the 5 nearest neighbors and their cosine distances are shown.

