# OpenReview forum: "RETVec: Resilient and Efficient Text Vectorizer"
_NeurIPS.cc/2023/Conference — NeurIPS 2023 poster_

### Official Review · Reviewer_LKoT · 2023-07-04

**Soundness:** 3 good
**Presentation:** 3 good
**Contribution:** 2 fair
**Rating:** 7
**Confidence:** 3

**Summary:**

This paper introduces RETVec, a resilient and multilingual text vectorizer designed for neural-based text processing. RETVec combines a unique character encoding with an optional small model to embed words into a 256-dimensional vector space. The RETVec embedding model is pre-trained using pair-wise metric learning, making it robust against typos. RETVec does not require dataset pre-processing and does not have out-of-vocabulary (OOV) tokens, as it accepts all valid UTF-8 characters. The authors provide a comprehensive evaluation of RETVec, demonstrating that it is faster and less memory-intensive than other vectorizers on multi-core CPUs and GPUs. Models trained with RETVec have slightly higher accuracy, greater resilience to typos, and better resilience to adversarial attacks compared to models trained with other vectorizers.

**Strengths:**

- The paper introduces RETVec, a text vectorizer that combines a unique character encoding with an optional small model, which addresses several challenges associated with existing text vectorizers.
- The paper provides a comprehensive evaluation of RETVec, demonstrating its performance in terms of speed, memory usage, and resilience to typos and adversarial attacks. Also, RETVec has the potential to significantly improve the performance of neural-based text processing, particularly in multilingual settings and in situations where typos and adversarial attacks are common.
- The authors provide code for their method, and they promise to make it open-source to the public, which will further facilitate the community.

**Weaknesses:**

- Though the proposed method exhibits significant improvements on several evaluations, it does not outperform sentencepiece overall when used for training pre-trained language model (BERT), which is a main-stream approach for many NLP tasks.
- It is vague how to effectively combine the proposed method with pre-trained language model, especially in light of the recent emergence of large language models. The paper would have been strengthened by discussing these aspects.

**Questions:**

Major:
1. In light of the recent advances in large language models (LLMs), it would be better to discuss the potential usage of the proposed method.
2. Regarding the experiments on pre-training BERT, it seems that the proposed method will change the traditional training routine of these pre-trained models. Unfortunately, the proposed method does not exhibit significant improvements over sentencepiece tokenizer. Besides the results in Figure 5, it would be better to illustrate the potential usage of the proposed method combined with pre-trained models (and LLMs, of course). Also, does the proposed method boost the training speed or convergence speed for BERT?

Minor:
1. line 53: Glove -> GloVe
2. line 240: missing reference to GLUE benchmark

**Limitations:**

A more explicit discussion of the limitations of RETVec would strengthen the paper and provide a more balanced view of its potential applications and implications. For example, a discussion on its inferior results on training BERT, and talk about what would be the best way or recommended way to use the porposed approach with BERT (or similar)?

---

> ### Author Rebuttal · Authors · 2023-08-09
>
> We thank the reviewer for their thoughtful review and constructive feedback. Please find our responses below.
>
> > **Q1:** Though the proposed method exhibits significant improvements on several evaluations, it does not outperform sentencepiece overall when used for training pre-trained language model (BERT)...
>
> **A1:**  Currently, we have not found a way to leverage RETVec to outperform other vectorizers across all tasks for pre-trained BERT models while providing increased robustness and efficiency. RETVec’s resilience comes from its learned embeddings which are pre-trained using a deep similarity loss, which means that the vocabulary size is not limited, as is in the case of standard tokenizers like SentencePiece. The major downside of not having a restricted set of token IDs in the vocabulary is that RETVec embeddings cannot be converted into a softmax output, which makes the standard pre-training tasks such as masked language modeling (predicting masked token IDs) unusable as-is. As a result, we needed to use alternative pre-training tasks to pre-train language models such as BERT. We felt that it was important to report this shortcoming as it emphasizes the current limitations for RETVec.
>
> We plan to use the extra space in the final version to discuss this challenge in more detail, describe interesting directions to explore in future work for combining RETVec with pre-trained language models, as well as add additional references that highlight the potential tradeoffs between robustness and accuracy [1, 2].
>
> [1] Zhang et al. “Theoretically Principled Trade-off between Robustness and Accuracy.” ICML, 2019.
>
> [2] Tsipras et al. “Robustness May Be at Odds with Accuracy.” ICLR, 2019.
>
> > **Q2:** It is vague how to effectively combine the proposed method with pre-trained language model...
>
> **A2:** We experimented with various approaches and reported the most successful one we found so far in the paper in Section 8. As discussed above, we don’t have a definitive answer yet on what is the best way to pre-train large language models using RETVec, especially since combining contrastively pre-trained word embeddings with LLMs is a largely unexplored area of study. We will make sure to include a discussion on this topic in the future work section, as we think the question deserves its own research given its complexity.
>
> > **Q3:** In light of the recent advances in large language models (LLMs), it would be better to discuss the potential usage of the proposed method.
>
> **A3:**  We will devote the extra space in the revised paper to address this point and emphasize that future work is needed to develop a better pre-training methodology that is more compatible with RETVec, due to the unique challenges of using pre-trained word embeddings instead of token IDs to represent text. As part of this discussion, we will make sure to highlight the potential benefits of using RETVec as the vectorizer for LLMs, including better multilingual capabilities, adversarial robustness, and smaller model size. In particular, for “small” or “medium-scale” LLMs (less than or around 1 billion total parameters), the embedding layers' parameters often accounts for more than 20% of the total parameters [3], which could be potentially reclaimed by using RETVec.
>
> [3] Biderman et al. “Pythia: A Suite for Analyzing Large Language Models.” arXiv:2304.01373.
>
> > **Q4:** Regarding the experiments on pre-training BERT, it seems that the proposed method will change the traditional training routine of these pre-trained models...it would be better to illustrate the potential usage of the proposed method combined with pre-trained models (and LLMs, of course).
>
> **A4:**  Pre-trained BERT with RETVec exhibits higher resilience to adversarial attacks and typos compared to SentencePiece (Figure 5) while remaining competitive (RETVec surpasses SentencePiece performance on 5/8 of the GLUE tasks, as shown in Table 5). As discussed above, we have not yet found the best pre-training methodology compatible with RETVec which will provide these benefits (efficiency and robustness) while offering equal or better performance on all tasks. Finding a more effective approach for pre-training RETVec-based models is the clear next step for our future work. For now, we reported this tradeoff and provided an in-the-wild study of the practical benefits of using RETVec with BERT when classifying adversarial text content such as spam (Section 9). We will use the extra space we have to discuss the usage and potential applications of RETVec in large pre-trained models including LLMs and our plans for future work.
>
> > **Q5:** Also, does the proposed method boost the training speed or convergence speed for BERT?
>
> **A5:** Yes, the training speed of BERT with RETVec is slightly faster than BERT with SentencePiece. The same number of training steps are needed so the convergence speed remains the same (the same number of training steps were also used to ensure a fair comparison in our benchmarks). We will make sure to highlight this fact in Section 8 and report overall pre-training and fine-tuning computational costs.
>
> > Minor: line 53: Glove -> GloVe; line 240: missing reference to GLUE benchmark
>
> Thank you for spotting those, we will correct them.
>
> > **Q6:** A more explicit discussion of the limitations of RETVec would strengthen the paper and provide a more balanced view of its potential applications and implications...
>
> **A6:**
> We thank the reviewer for their insightful comments, and recognize the importance of such discussion. We will make sure to devote the extra space in the final version to discuss limitations and challenges in the existing pre-training methodology for BERT, as well as potential applications and future work to adapt RETVec to large pre-trained language models.
>
> We hope that our responses helped address your questions. Please let us know if you have any further questions or feedback. Thank you for the review!

---

> > ### Comment · Reviewer_LKoT · 2023-08-21
> >
> > Thanks for your response. I slightly increased my rating to reflect the authors' response.

---

### Official Review · Reviewer_AFvP · 2023-07-05

**Soundness:** 3 good
**Presentation:** 3 good
**Contribution:** 3 good
**Rating:** 7
**Confidence:** 2

**Summary:**

This paper introduces RETVec, a resilient and efficient text vectorizer designed for neural-based text processing. RETVec is a multilingual tool that combines a novel character encoding with a pre-trained embedding model to create a 256-dimensional vector space. The vectorizer is significantly more resilient to typos and adversarial text attacks than other state-of-the-art vectorizers.The results show that RETVec outperforms other vectorizers in terms of accuracy and resilience to typos and adversarial attacks.

**Strengths:**

RETVec is a novel and efficient text vectorizer that is designed to be resilient to typos and adversarial text attacks. This is a significant contribution to the field of natural language processing, as text vectorization is a critical component of many NLP tasks.

The paper provides a detailed description of RETVec's architecture and evaluation methodology. This makes it easier for other researchers to understand and replicate the results of the paper.

The paper evaluates RETVec's performance on four different datasets with drastically different dataset sizes, number of languages, classification tasks, and text lengths. This demonstrates the versatility and effectiveness of RETVec across a wide range of NLP tasks and datasets.

The results show that RETVec outperforms other vectorizers in terms of accuracy and resilience to typos and adversarial attacks. This is a significant finding, as it suggests that RETVec could be a valuable tool for real-world NLP applications where robustness to errors and attacks is critical.

**Weaknesses:**

The paper does not provide a detailed analysis of RETVec's limitations and potential failure cases. While the paper does mention some of the challenges faced during the development of RETVec, a more thorough analysis of its limitations and potential failure cases would have been helpful to better understand the scope of its applicability.

The paper conducts experiments on classification tasks. I'm curious how it performs on text generation tasks, like machine translation. Since this determines the scalibility of a practical text vectorizer.

**Questions:**

Please refer to the weakness

**Limitations:**

Yes

---

> ### Author Rebuttal · Authors · 2023-08-09
>
> We thank the reviewer for their thoughtful review and insightful comments. Please find our response below.
>
> > **Q1:** The paper does not provide a detailed analysis of RETVec's limitations and potential failure cases. While the paper does mention some of the challenges faced during the development of RETVec, a more thorough analysis of its limitations and potential failure cases would have been helpful to better understand the scope of its applicability. The paper conducts experiments on classification tasks. I'm curious how it performs on text generation tasks, like machine translation. Since this determines the scalibility of a practical text vectorizer.
>
> **A1:** Through this review, we realized that we have been overly focused on classification tasks and adversarial resilience, which were the main goals of RETVec. We plan on fixing this in the revised version by adding an in-depth discussion on the challenges and potential approaches for applying RETVec to other tasks, including text generation tasks, other sequence-to-sequence tasks, and pre-training large language models. Generative tasks are challenging because the 256-float embedding returned by RETVec cannot be converted into a softmax output like in the case of token IDs outputted by other vectorizers such as SentencePiece. As a result, there is no straightforward way of training a generative model which predicts the next token ID in the sequence, and we have to experiment with alternative forms of pre-training such as predicting the top N words, training a decoder for the RETVec embedding model, decoding character-by-character, or using a VQ-VAE model [1]. There has been limited work on the area of combining contrastively pre-trained word embeddings with pre-trained language models on text generation tasks. Thus, we are still unsure which methodology is the best, and we plan on using the extra page in the final version to discuss these challenges, potential applications, our initial results, and outline the need for future work in this direction.
>
> Please let us know if you have any additional questions or feedback. Thank you for the review!
>
> [1] Van den Oord et al. “Neural Discrete Representation Learning.” NIPS 2017. arXiv:1711.00937.

---

> > ### Comment · Reviewer_AFvP · 2023-08-20
> >
> > Thanks for your response, and it resolves my concerns! I decide to raise the score.

---

### Official Review · Reviewer_aZHy · 2023-07-07

**Soundness:** 3 good
**Presentation:** 2 fair
**Contribution:** 3 good
**Rating:** 7
**Confidence:** 4

**Summary:**

The paper introduces RETVec, a resilient, efficient, and multilingual text vectorizer designed for neural-based text processing. It addresses the limitations of existing approaches by combining a novel UTF-8 character encoder with a small model. RETVec does not require dataset pre-processing and accepts all valid UTF-8 characters, eliminating the need for out-of-vocabulary tokens. The embeddings are trained using pair-wise metric learning, ensuring that words with typos are embedded close to the original word. RETVec outperforms other vectorizers on text classification tasks, exhibiting higher accuracy, greater resilience to typos, and better resilience to adversarial attacks. The paper provides a TensorFlow implementation of RETVec, along with pre-trained models.

**Strengths:**

1) Addresses the limitations of existing text vectorization approaches.
2) Combines a novel UTF-8 character encoder with a small model.
3) Does not require dataset pre-processing and eliminates the need for out-of-vocabulary tokens.
4) Trained on a word dataset with more than 157 languages.
5) Space-efficient and suitable for on-device model deployment.
6) Outperforms other vectorizers on text classification tasks, with improved accuracy and resilience to typos and adversarial attacks.
7) Provides a TensorFlow implementation and pre-trained models.

**Weaknesses:**

The paper does not provide detailed comparison results with other vectorizers on different languages and multilingual settings.

**Questions:**

1) How does the character encoder handle rare or uncommon characters in the UTF-8 character set?
2) Are there any limitations or performance trade-offs when using RETVec with extremely long words?

**Limitations:**

1) The paper focuses on text classification tasks and does not explore other natural language processing tasks.
2) The paper does not provide insights into the interpretability of RETVec embeddings and their usefulness in downstream tasks.

---

> ### Author Rebuttal · Authors · 2023-08-09
>
> We thank the reviewer for their thoughtful review and constructive feedback. Please find our responses below.
>
> > **Q1:** The paper does not provide detailed comparison results with other vectorizers on different languages and multilingual settings.
>
> **A1:** We evaluated RETVec on the Amazon Multilingual Reviews Corpus and reported the results in Figure 2, broken down per language. Figure 2 shows that RETVec outperforms the four baseline text vectorizers including SentencePiece and BPE on all 6 languages in the dataset, with a strong lead in Chinese and Japanese. We will add results on another dataset with more languages for the revised version.
>
> > **Q2:** How does the character encoder handle rare or uncommon characters in the UTF-8 character set?
>
> **A2:** RETVec’s character encoder uses 24 bits per character. The character encoder converts every valid UTF-8 character into its integer codepoint before converting it into a binary representation, which ensures that 100% of the UTF-8 character set can be uniquely represented. Additionally, for the RETVec model, we ensure that all UTF-8 characters are seen during training by including 10% random UTF-8 character strings in the training dataset and applying random character insertion and substitution augmentations.
>
> In order to visualize how RETVec handles uncommon words with rare UTF-8 characters, we will add a plot of the similarity distance between words and their typo-laden versions for both a set of common words and a set of random strings/uncommon words using the same typos, and show that they are comparable. This will demonstrate that every token, including those containing rare UTF-8 characters, is handled in a similar fashion by RETVec.
>
> > **Q3:** Are there any limitations or performance trade-offs when using RETVec with extremely long words?
>
> **A3:** In the ablation study, we trained RETVec models with input word lengths ranging from 12 to 32, but we did not see any performance improvements by increasing the word length above 16 characters per word (Table 6). Furthermore, increasing the input word length also increases RETVec’s model size and latency – we will add these metrics to Table 6 as well and discuss them in further detail in Section 10.
>
> > **Q4:** The paper focuses on text classification tasks and does not explore other natural language processing tasks.
>
> **A4:** RETVec is designed for adversarially resilient text classification and with on-device use-cases in mind, which is why most of our benchmarks are focused around classification performance and adversarial robustness. We realized that we should have devoted more space to discuss other use-cases such as text generation in future work and plan to use the extra space in the revised version to correct this.
>
> > **Q5:**  The paper does not provide insights into the interpretability of RETVec embeddings and their usefulness in downstream tasks.
>
> **A5:**  We will try to display the clusters of word embeddings using an embedding projector (https://projector.tensorflow.org/) and add it to the GitHub repository and paper appendix. This visualization will help demonstrate that syntactically similar words (e.g. a word and a typo version of a word) are clustered closer together while the embeddings of different and dissimilar words are further apart. This will hopefully provide some intuition on the RETVec embedding space and offer insights into the interpretability of RETVec embeddings. We are unsure how to provide insights into the embeddings’ usefulness in downstream tasks.
>
>
>
> Please let us know if you have any additional questions or feedback, we would be happy to incorporate any further feedback into the revision of the paper. Thank you!

---

### Comment · Area_Chair_fR8x · 2023-08-18
**discussion**

Dear Reviewers,

Thanks for your initial reviews. Could you please acknowledge authors' responses and engage in a discussion (if needed)?

Thanks
AC

---

### Decision · Program_Chairs · 2023-09-21

**Decision:**

Accept (poster)

**Comment:**

The proposed RETVec is a robust and effective multilingual text vectorization method, as acknowledged by all the reviewers. Reviewers also appreciated the authors' response in clarifying some questions. I'm in favor of accepting the paper